# Longitudinal variability in the urinary microbiota of healthy premenopausal women and the relation to neighboring microbial communities: A pilot study

Lena M. Biehl[1,2], Fedja Farowski[1,2,3], Catharina Hilpert[1], Angela Nowag[4,5], Anne Kretzschmar[4], Nathalie Jazmati[4,5], Anastasia Tsakmaklis[1], Imke Wieters[3], Yascha Khodamoradi[3], Hilmar Wisplinghoff[4,5,6], Maria J. G. T. Vehreschild[1,2,3]*

**1** Faculty of Medicine and University Hospital of Cologne, Department I of Internal Medicine, University of Cologne, Cologne, Germany, **2** German Centre for Infection Research, Bonn-Cologne, Germany, **3** Department of Internal Medicine, Infectious Diseases, University Hospital Frankfurt, Goethe University Frankfurt, Frankfurt am Main, Germany, **4** Wisplinghoff laboratories, Cologne, Germany, **5** Institute for Medical Microbiology, Immunology and Hygiene, University of Cologne, Cologne, Germany, **6** Institute for Virology and Medical Microbiology, Witten/Herdecke University, Witten, Germany

* maria.vehreschild@kgu.de

**Data Availability Statement:** The dataset generated and analyzed in this study is available in

## Abstract

### Background

The understanding of longitudinal changes in the urinary microbiota of healthy women and its relation to intestinal microbiota is limited.

### Methods

From a cohort of 15 premenopausal women without known urogenital disease or current symptoms, we collected catheter urine (CU), vaginal and periurethral swabs, and fecal samples on four visits over six months. Additionally, ten participants provided CU and midstream urine (MU) to assess comparability. Urine was subjected to expanded culture. 16S rRNA gene sequencing was performed on all urine, fecal, and selected vaginal and periurethral samples. Sequence reads were processed (DADA2 pipeline) and analyzed using QIIME 2 and R.

### Results

Relative abundances of urinary microbiota were variable over 6–18 months. The degree of intraindividual variability of urinary microbiota was higher than that found in fecal samples. Still, nearly half of the observed beta diversity of all urine samples could be attributed to differences between volunteers ($R^2 = 0.48$, $p = 0.001$). After stratification by volunteer, time since last sexual intercourse was shown to be a factor significantly contributing to beta diversity ($R^2 = 0.14$, $p = 0.001$). We observed a close relatedness of urogenital microbial habitats and a clear distinction from intestinal microbiota in the overall betadiversity analysis. Microbiota compositions derived from MU differed only slightly from CU compositions. Within this

the NCBI Sequence Read Archive under the BioProject accession number PRJNA649069.

**Funding:** This study was supported by a study grant from the German Research Foundation (DFG - https://www.dfg.de/), grant number BI 1899/1-1, awarded to Lena M. Biehl. The funder had no role in study design, data collection and analysis, decision to publish, or preparation of the manuscript.

**Competing interests:** I have read the journal's policy and the authors of this manuscript have the following competing interests: LMB has received lecture honoraria from Astellas and Merck/MSD, and travel grants from 3M and Gilead. MJGTV reports grants and personal fees from 3M, Alb Fils Kliniken GmbH, Astellas Pharma, Basilea, bioMérieux, DaVolterra, Gilead Sciences, Ferring, Glycom, Heel, MaaT Pharma, Merck/MSD, Organobalance, Pfizer, Roche Pharma, Seres Therapeutics. YK has received lecture honoraria from Merck/MSD and Gilead, and travel grants from Gilead. All remaining authors have declared no conflicts of interest. This does not alter our adherence to PLOS ONE policies on sharing data and materials.

analysis of low-biomass samples, we identified contaminating sequences potentially stemming from sequencing reagents.

## Conclusions

Results from our longitudinal cohort study confirmed the presence of a rather variable individual urinary microbiota in premenopausal women. These findings from catheter urine complement previous observations on temporal dynamics in voided urine. The higher intraindividual variability of urinary microbiota as compared to fecal microbiota will be a challenge for future studies investigating associations with urogenital diseases and aiming at identifying pathogenic microbiota signatures.

## Introduction

For many decades, the urinary tract of healthy individuals was considered to be sterile. This perception was based on a lack of cultivable microorganisms in urine samples obtained by clean-catch urine or catheterization [1]. Contrary to this former dogma, the existence of a specific bladder microbiota has now been widely recognized [2,3]. Both, expanded urinary culture and 16S rRNA gene sequencing have proven that a microbial community exists in the bladder of healthy women [4–10]. Consequently, some studies have started to investigate its role in certain urogenital conditions [11–13]. Yet, knowledge about physiological attributes of urinary microbiota as a prerequisite for studying specific alterations and dynamics associated with diseases is still limited [14]. In particular, evidence on intraindividual longitudinal changes in the urinary microbiota is extremely scarce [15–17]. The only available data on temporal changes of the urinary microbiota in premenopausal women stems from a very recent study with daily sampling of voided urine from eight participants over three months [16]. This detailed investigation showed quite dynamic urinary microbiota with intermittent changes and potential influence by sexual activity and menstruation. Still, the microbiota compositions as determined by expanded quantitative urine culture (EQUC), displayed some degree of resilience or stability by returning to former states. In order to further characterize the stability of healthy urinary microbiota over time and compare this with the stability of intestinal microbiota as the most well-studied microbial community in humans, complementary evidence is needed. In this context, assessment of anatomically closely related microbial communities in the vagina, periurethral skin, and intestine provides important insights into the interconnectedness of these habitats [6,18–21]. Lastly, as samples with low bacterial abundance, urine specimens are prone to contamination, in particular when employing voided urine rather than catheterized [17]. However, transurethral catheterization is more invasive and bears risks. While both sampling techniques were applied in previous studies on urinary microbiota, only few studies directly compared microbiota analysis derived from transurethral catheterization to those from midstream urine samples from identical patients or volunteers [6,22,23]. Since both reservoirs may influence urogenital disorders, it seems important to characterize previously described differences also in female asymptomatic volunteers. Thus, we established a cohort of healthy female volunteers investigating the longitudinal variability of urinary microbiota, its relation to vaginal, periurethral and fecal microbiota, and the impact of different urine sampling techniques.

## Materials and methods

### Female volunteers and data collection

The study was conducted at the University Hospital of Cologne, Germany. Pre-menopausal adult women without known urogenital disease or current presence of urogenital symptoms were eligible for inclusion in the study. Exclusion criteria were current pregnancy, urinary tract infection within six months and intake of antibiotic substances within three months prior to study inclusion. Due to the exploratory character of this pilot study, inclusion of 10–15 participants was planned without prior statistical sample size calculation.

Recorded data included medical history, diet (non-vegetarian, lactose-free, vegetarian), average daily fluid intake (<1.5 l/d, 1.5–3 l/d, >3 l/d), and for each sampling visit current medication including hormonal contraception, week of menstrual cycle (1 to 4), time since last sexual intercourse (<24 h, 24–96 h, >96 h—1 week, >1 week), current urogenital symptoms and symptoms associated with sampling procedures.

The study was approved by the responsible ethics committee (UKK 15–314) and written informed consent of all participants was obtained. It was conducted in accordance with the Declaration of Helsinki.

### Sample collection

At baseline, two, four and six months after inclusion, catheter urine (CU) using disposable catheters (SpeediCath CH10 Nelaton Kinder, Coloplast GmbH, Hamburg, Germany), vaginal swabs (vswab) and periurethral swabs (pswab; both eSwab containing Aimies medium, COPAN ITALIA S.P.A., Brescia, Italy) were collected (visit 1–4). Prior to collection of pswabs and CU, the external urethral ostium and surrounding skin were disinfected. Furthermore, all participants provided a fresh fecal sample on the day of each sampling visit or within four days using cooled transport (Fridge-to-go, Playtex Baby, New York, USA). Timing of sample visits was regardless of exact week in menstrual cycle, but sampling did not take place during menstruation. As an additional substudy assessing sample comparability, a fifth visit was initiated one year after the fourth visit. On this occasion, midstream urine (MU) was collected immediately after collection of CU, which was performed without complete emptying of the bladder.

If participants developed a urinary tract infection (UTI) during the study, additional sample collection visits were performed before initiation of antibiotic treatment and 2–5 days afterwards.

Samples were labeled with the participant ID (letters A-O), the visit (1–5 or UTI1+2) and the material and transported under cooled conditions (4–8°C). Samples from each specimen were divided into three equal aliquots for further analysis. Within four hours of sample collection, one aliquot (5–8 ml) of all urine samples was subjected to expanded urinary culture. All remaining aliquots were stored at -80°C until further processing.

In addition to participant samples, negative sampling controls of sterile saline (0.9%) applying the same sample catheters, urine collection devices and procession steps as previously described for the volunteer samples. Furthermore, negative and positive extraction and PCR controls were included in the consecutive sequencing.

### Expanded urinary culture

Urine samples were streaked out with a 1 µl inoculation loop on CPS Elite agar (CPSE), tryptic soy (TSA) agar, chocolate agar, and Schaedler Kanamycin-Vancomycin (SKV) agar. CPSE, TSA, and chocolate agar plates were incubated for 18–24 hours at 37°C with 5% CO2. SKV plates were incubated anaerobically at 37°C.

In addition, 1 ml of each urine sample was inoculated in 9 ml thioglycollate broth for 14 days at 37˚C with 5% CO2. In case of turbidity, CPSE, TSA, SKV, and chocolate agar plates were inoculated with a 1μl inoculation loop and incubated for 18–24 hours at 37˚C either with 5% CO2 or anaerobically (SKV).

In case of bacterial growth, colony-forming units (CFU) were counted and calculated to CFU/ml. Bacterial identification was performed using matrix-assisted laser desorption/ionization time-of-flight mass spectrometry (MALDI-TOF).

## DNA extraction

Genomic DNA of each urine sample was isolated using the UltraClean Microbial DNA Isolation kit (MoBio Laboratories, Inc., Carlsbad, USA). Briefly, aliquots of 5–8 ml were centrifuged (20,800g, 15 min) and the resulting pellets were suspended in 350 μl MicroBead Solution and an enzyme digestion with 40 μl Lysozyme (50 mg/ml), 16 μl Mutanolysin (10 U/μl) and 16 μl RNaseA (10 U/μl; all Sigma-Aldrich, St. Louis, USA) was performed for 45 min at 37˚C according to published protocols [24]. For cell lysis, 40 μl Proteinase K (20 mg/ml; Sigma-Aldrich, St. Louis, USA) and 60 μl MD1 solution from the MoBio kit were added and the solution was incubated for 10 min at 65˚C. Further DNA purification steps were performed in accordance with the manufacturers' instructions.

To isolate the DNA from vswabs and pswabs, 200 μl swab medium was used for an enzyme digestion with 200 μl Lysozym (5 mg/ml) 20 μl Mutanolysin (10 U/μl) and 20 μl RNaseA (10 U/μl; all Sigma-Aldrich, St. Louis, USA) at 65˚C for 10 min. Following this initial digestion, the QIAamp DNA Mini Kit (Qiagen, Hilden, Germany) was used by adding 50 μl Proteinase K (20 mg/ml; Sigma-Aldrich, St. Louis, USA) and 200 μl Qiagen AL buffer (Qiagen, Hilden, Germany) followed by incubation for 10 min at 56˚C. Further DNA purification steps were performed following manufacturers' instructions.

From stool samples, total DNA was isolated using the RNeasy Power Microbiome Kit (Qiagen, Hilden, Germany) skipping steps 11 to 13 according to manufacturers' instructions.

## Sequencing

Taxonomic profiling was performed by amplicon sequencing of the V3-V4 region of the bacterial 16S rRNA gene. Briefly: The 16S V3-V4 region was amplified (25 cycles; template: up to 12.5 ng gDNA; KAPA HiFi HotStart Ready Mix) using the primers 341F and 802R (0.2 μM each) as published elsewhere [25]. This amplicon was then purified using the Agencourt AMPure XP PCR Purification system (Beckman Coulter, Krefeld, Germany), processed (indexed, purified, normalized and pooled) and sequenced in a 300 bp paired-end run using the MiSeq Reagent Kit v3 (Illumina, San Diego, California) as outlined in the Illumina 16S Sample Preparation Guide [26].

Since the concentration of the extracted DNA from urinary, vaginal, and periurethral samples was low, the volume of the template for the first (i.e. amplicon) PCR was increased to 10.5 μl; the final volume (25 μL) and concentrations remained unchanged. The presence and concentration of the respective amplicons were confirmed using the Bioanalyzer (Agilent Genomics, Santa Clara, USA).

## Data processing and data analysis

Sequencing data was processed using the DADA2 pipeline and QIIME 2 [27,28]. Quality profiles of the reads were analyzed, trimmed (trunc_len_f = 300, trunc_len_r = 260) and processed using the QIIME 2 DADA2 plugin with the *denoise-paired* option and standard parameters (trunc_q = 2, max_ee = 2, chimera_method = consensus). Taxonomic classification was done

by a Naïve Bayes classifier (sklearn) [29], trained on SILVA database release 138 [30]. In order to eliminate sequences originating from contamination during sampling and processing steps, we filtered amplicon sequence variants (ASVs) with the following criteria: mean abundance in negative controls over 0.5%, presence of ASV in all negative sampling controls and relative abundance of ASV >5% in at least one negative sampling control. These ASVs were removed from the feature table. Analysis of the relative proportion of each bacterial taxon was performed after the feature table was rarefied at a depth of 2,000 sequences per sample.

Statistical analyses were carried out using R for Statistical Computing (version 3.2.5, R Foundation for Statistical Computing, Vienna, Austria) [31]. The QIIME 2 data was imported and diversity scores were calculated using the phyloseq R package [32]. The beta diversity, in this case the generalized UniFrac distances between the samples, was visualized using principal coordinate analysis (PCoA) and the effect of volunteer variables on the beta diversity was tested by a permutational multivariate analysis of variance (PERMANOVA). Microbiota stability was assessed using Jensen-Shannon divergence (JSD) calculated on ASV level. JSD was used to measure similarity between two probability distributions. A lower JSD value represents higher similarity between compared samples. For each sample, the median JSD between the respective sample and all other samples in the same group/by the same volunteer was determined.

All continuous data was presented as mean and standard deviation (SD) or median and range, presented as box plots and tested with Mann-Whitney U-test and Kruskal-Wallis-test with Dunn's post test, as appropriate. Only genera with a mean relative abundance >1% were used for comparisons between sample groups. All statistical tests were two-tailed, and a corrected p-value below 0.05 was considered statistically significant.

Taxonomic classification at the genus level was primarily used to compare results of 16S rRNA gene sequencing with those of expanded culture.

## Results

### Cohort characteristics

Between November 2016 and January 2017, 15 premenopausal female volunteers with a median age of 23 years (range 20–32) were included in the study. Overall, nine (60.0%) volunteers reported previous UTIs in their medical history; no other urogenital diseases were reported. Table 1 outlines the characteristics of the cohort.

### Sample collection

All 15 volunteers participated in four regular sample visits within six months each. Volunteer F did not undergo catheterization on the fourth sampling visit due to light pelvic pain and macro hematuria immediately after the third sampling visit. In volunteer O, catheterization was not successful at the third sampling visit. Volunteer K developed a UTI requiring antibiotic treatment between her third and fourth sampling visits. She provided MU on the day of diagnosis (UTI1) and CU, fecal, and swabs three days afterwards (UTI2). Results of these two samples were not included in the overall analysis of study samples. Her fourth sampling visit (CU4) was obtained 35 days after the antibiotic treatment. Three volunteers (C, D, E) developed an infection not affecting the urogenital tract during the course of the study and received antibiotic treatment between two sample visits. In total, six sampling visits took place after antibiotic exposure, but only three of these with a more recent antibiotic exposure within 40 days prior sampling (C4CU 40 days after antibiotic exposure, D3CU 16 days and K4CU 35 days).

**Table 1. Characteristics of the study cohort.**

| Variable | Cohort (N = 15) |
|---|---|
| Median Age in years (range) | 23 (20–32) |
| Urological diseases (%) | |
| 1 previous UTI | 3 (20.0) |
| >1 previous UTI | 6 (40.0) |
| Other urogenital diseases | 0 |
| Abdominal/urogenital surgery in history (%)[a] | 2 (13.3) |
| Any underlying disease[b] (%) | 4 (26.7) |
| Regular medication (%) | |
| Oral contraceptives | 8 (53.3) |
| Other[c] | 3 (20.0) |
| Diet (%) | |
| Non-vegetarian | 11 (73.3) |
| Non-vegetarian, lactose free | 2 (13.3) |
| Vegetarian | 2 (13.3) |
| Reported average fluid intake (%) | |
| < 1.5 l/day | 2 (13.3) |
| 1.5–3.0 l/day | 10 (66.7) |
| > 3.0 l/day | 3 (20.0) |

UTI: Urogenital tract infection.

[a] Inguinal hernia surgery (both) and postpartal curettage (1).

[b] Hypothyreosis on substitution (3), Fibromyalgia and chronic gastritis (1), Pendred syndrome (1), Newly diagnosed Multiple sclerosis without any symptoms or signs of neurogenic bladder (1).

[c] Levothyroxin for hypothyreosis (3), Interferon-ß for Multiple sclerosis (1).

A fifth sampling visit including CU and MU was conducted in a subgroup of ten volunteers in June 2018. In summary, 69 CU, 11 MU, 61 fecal samples, 59 pswabs and 60 vswabs were collected from the study cohort.

Catheterization was well tolerated in nearly all cases. In addition to above described adverse effects, five (33.3%) volunteers reported a slight burning sensation during the disinfection, possibly related to earlier shaving.

## Detected taxa by sequencing and comparison with expanded culture

DNA extraction of CU samples yielded a mean DNA concentration of 2.34 ng/µl (standard deviation [SD] ±4.36, minimum 0.13, maximum 27.84). 16S rRNA gene sequencing resulted in mean number of reads of 51,461 (SD ±32,589). Six ASVs belonging to 4 genera were removed from the dataset due to their high abundance in the negative sampling controls (Alkalibacterium, Bacteroides, Shewanella, and Serratia, see S1 Fig) with Bacteroides being the most abundant one among these. Their abundance in volunteers' samples correlated negatively with the DNA concentration after extraction (Pearson's product-moment correlation -0.601; [95%-CI 0.785–0.319]) indicating a contamination at one point between sample collection and sequencing. Removing these ASVs lead to a substantial reduction in sequence reads in CU samples (mean number of reads 23,394±24,471). After application of a rarefaction threshold of 2,000 reads, five CU samples yielded too few reads and were excluded from the analysis (H1CU, D5CU, L5CU, I5CU, B5CU). Microbiota analysis revealed the presence of bacterial DNA in the remaining samples (63/68; 92.6%). The most abundant taxon on the

genus level was *Lactobacillus* (49.7%±37.7; 61/63 samples). See Fig 1A and S1 Table in the Supporting Information for details on relative abundances.

Sixty-seven CU samples were subjected to expanded culture, one sample was not cultured due to limited sample volume (F3CU). Eight (11.9%) samples yielded no bacterial growth, 23 (34.3%) yielded one bacterial species, 23 (34.8%) two bacterial species, and 13 (19.4%) three or more species. The most frequently detected genera were *Lactobacillus* (43/67 samples; 64.2%) followed by *Gardnerella* (15/67 samples; 22.4%). S2 Table shows detailed results of expanded culture of CU samples.

Results of both, 16S rRNA gene sequencing and expanded culture were available for comparison in 63 CU samples. In 44/63 (69.8%) cases, all cultured taxa were also detected by 16S rRNA gene sequencing; in 21 of these cases, the relative abundances of all cultured taxa were over 10%. In 10/63 (15.9%) cases, not all cultured taxa were confirmed in the sequencing results. No concordance between culture and sequencing results was seen in eight (12.1%) cases without any bacterial growth (S3 Table). The most frequently cultured genus *Lactobacillus* was confirmed by sequencing in 40 of 41 (97.6%) respective samples.

## Longitudinal changes of urinary microbiota compared to fecal microbiota

During the observation period of 6–18 months intra-individual changes in the alpha-diversity of urinary microbiota were observed (S2 Fig). Furthermore, the microbiota composition as seen in the relative abundance of taxa varied throughout the study for most volunteers (Fig 1A).

To further illustrate longitudinal variability, JSD values (a measure of divergence between samples) of CU samples 1–4 of each participant were put into context with respective values of fecal samples collected in parallel (Fig 1B). The median JSD value in CU samples was 0.26 (range 0.02–0.64; interquartile range [IQR] 0.14–0.37), which was significantly higher than that of fecal samples with 0.20 (range 0.10–0.57; IQR 0.12–0.19; p = 0.029). The larger IQR in CU samples indicates less homogeneity in the variability between different volunteers. In some volunteers, the longitudinal variability according to JSD values was rather small (in particular G, H, I, M, N), while others seemed to have less stable urinary microbiota compositions (B, F, O).

In a permutational multivariate analysis of variance using distance matrices (PERMANOVA), categorization by volunteer ID accounted for nearly half of the observed beta diversity of CU samples from visits 1–4 ($R^2$ = 0.48, p = 0.001). See Fig 1C for PCoA displaying beta diversity of CU samples 1–4 using generalized UniFrac. When including the more delayed CU samples from visit 5, volunteer ID remained a significant factor ($R^2$ = 0.44, p = 0.001). Of note, the proportion of compositional differences related to volunteer ID in fecal samples (visit 1–4) was higher ($R^2$ = 0.59, p = 0.001).

## Influence factors on urinary microbiota compositions

We then assessed the impact of variables potentially influencing beta diversity of CU samples 1–4 by PERMANOVA. In a univariate analysis, daily fluid intake and intake of a contraceptive pill were identified as factors impacting beta diversity (Table 2). However, these variables strongly depended on individual volunteer ID. Thus, they were not included in the multivariate approach. In the multivariate model, only volunteer ID ($R^2$ = 0.48, p = 0.001) and time since last sexual intercourse ($R^2$ = 0.09; p = 0.001) remained significant factors explaining compositional differences. Since most of the other variables were linked to volunteer ID, we repeated the analysis after stratification by volunteer ID. After stratification, time since last sexual intercourse ($R^2$ = 0.14, p = 0.001) and daily fluid intake ($R^2$ = 0.14, p = 0.014) were shown to significantly impact the composition of the urinary microbiota. See Table 2 for details.

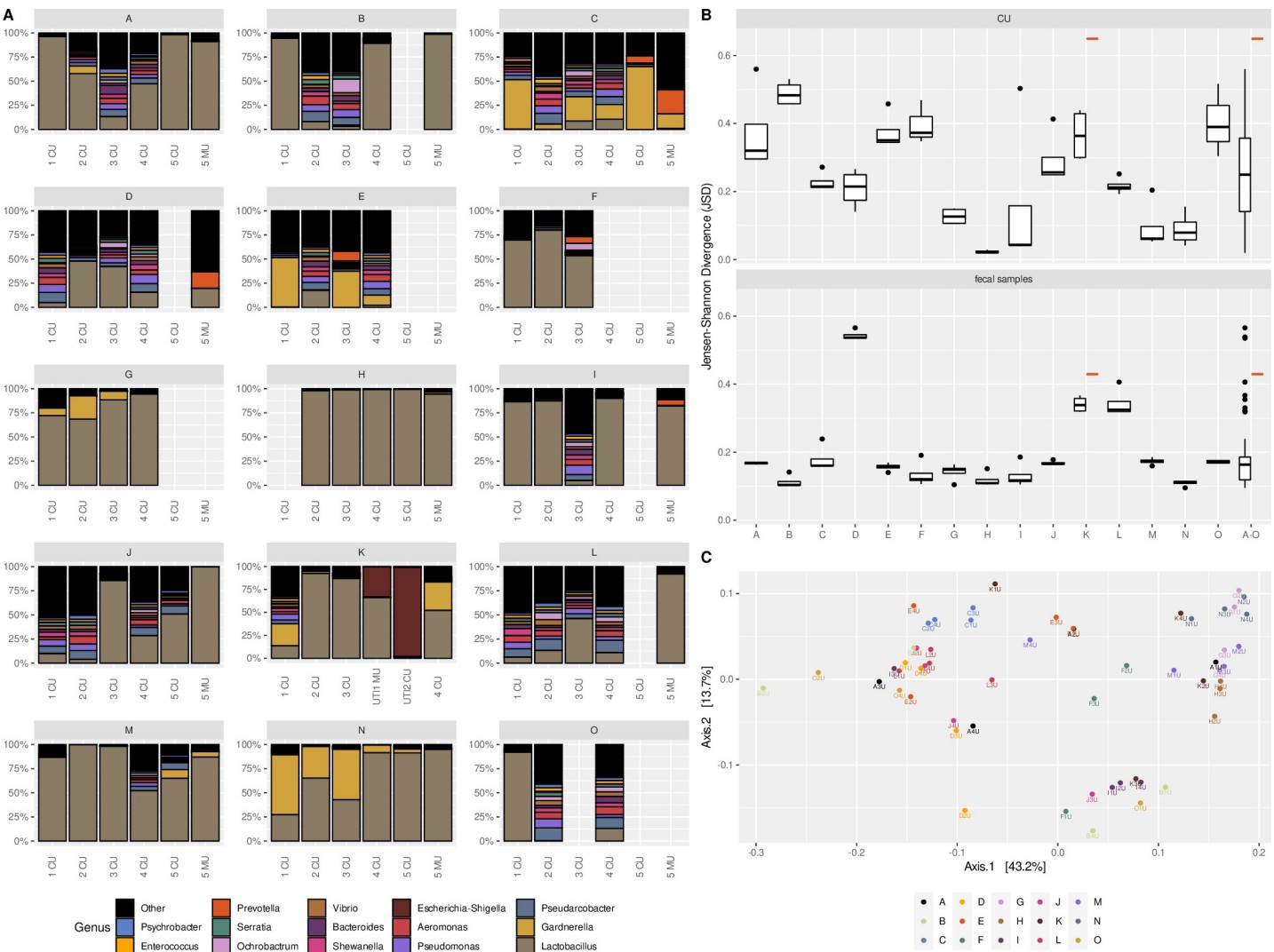

**Fig 1. Longitudinal variability of urinary microbiota. A: Relative abundances of bacterial genera in urine samples over time.** Letters A-O indicate individual volunteers. CU was collected at visits 1–4 (0, 2, 4 and 6 month); at visit 5 (18 months after baseline) CU and MU was collected from 10 subjects. Volunteer K developed a UTI between the third and fourth sampling visit, samples taken on the day of diagnosis, and three days later are indicated by UTI1MU and UTI2CU, respectively. Antibiotic exposure was reported 40 days prior sample C4CU, 16 and 63 days prior sample D3CU and D4CU, respectively, 89 days prior sample E4CU, and 35 days prior sample K4CU. **B: Longitudinal variability of CU and fecal samples from visit 1–4 shown as median Jensen-Shannon Divergence (JSD) values per volunteer.** JSD calculations were based on amplicon sequence variants (ASV) level. The red dashes represent samples UTI2CU and the corresponding fecal sample from volunteer K collected three days after diagnosis and antibiotic treatment of a UTI. The plot "A-O" represents the median of all intraindividual JSD values. **C: Beta diversity (generalized UniFrac) of CU samples from visit 1–4 visualized by principal coordinates analysis (PCoA).** Letters and color indicate volunteers. CU: Catheter urine; MU: Midstream urine; UTI: Urogenital tract infection.

## Neighboring microbial habitats

In addition to urine samples, the first pswab (N = 15), the first and fourth vswab (N = 30) and all fecal samples (N = 61) of each volunteer were subjected to 16S rRNA gene sequencing. One fecal sample (A3S) did not yield enough reads for further analysis. The by far most abundant taxon in both, vswabs and pswabs, was *Lactobacillus* (mean abundance in vswabs: 84.4% ± 28.7%; pswabs: 69.1% ± 34.5%) followed by *Gardnerella*. See S1 Table and Fig 2A for the most abundant taxa detected per sample type. Alpha diversity metrics showed the highest alpha diversity for fecal samples (mean Shannon index 3.91; SD 0.35) followed by CU (mean

**Table 2. Impact of volunteer variables on beta diversity (generalized UniFrac) of CU samples 1–4 by permutational multivariate analysis of variance using distance matrices (PERMANOVA).**

| Variable | Univariate | | Multivariate including ID[a] | | Multivariate stratified by ID | |
|---|---|---|---|---|---|---|
| | $R^2$ | p value | $R^2$ | p value | $R^2$ | p value |
| Diet | 0.05 | 0.128 | - | - | 0.08 | 0.005 |
| Daily fluid intake | 0.14 | 0.001 | - | - | 0.14 | 0.005 |
| Intake of contraceptive pill | 0.05 | 0.016 | - | - | 0.02 | 0.005 |
| Week of menstrual cycle | 0.09 | 0.123 | 0.04 | 0.457 | 0.06 | 0.607 |
| Time since last sexual intercourse | 0.08 | 0.132 | 0.09 | 0.004 | 0.14 | 0.001 |
| Volunteer ID[b] | 0.48 | 0.001 | 0.48 | 0.001 | - | - |

CU: Catheter urine.

[a]Model after excluding variables with nearly 100% consistency among samples of one volunteer: Diet, daily fluid intake and intake of contraceptive pill.

[b]Samples belonging to the same volunteer were grouped in one category.

Shannon index 2.27; SD 1.34, Table 3). Vswabs yielded the lowest alpha diversity (mean Shannon index 0.65; SD 0.60). PCoA illustrating beta diversity of all analyzed samples revealed a clear distinction between fecal samples and other sample types (Fig 2B). Within urogenital samples, microbiota from CU samples and vswabs seem to be most distinct based on gUniFrac distances (see S3 Fig for PCoA excluding fecal samples).

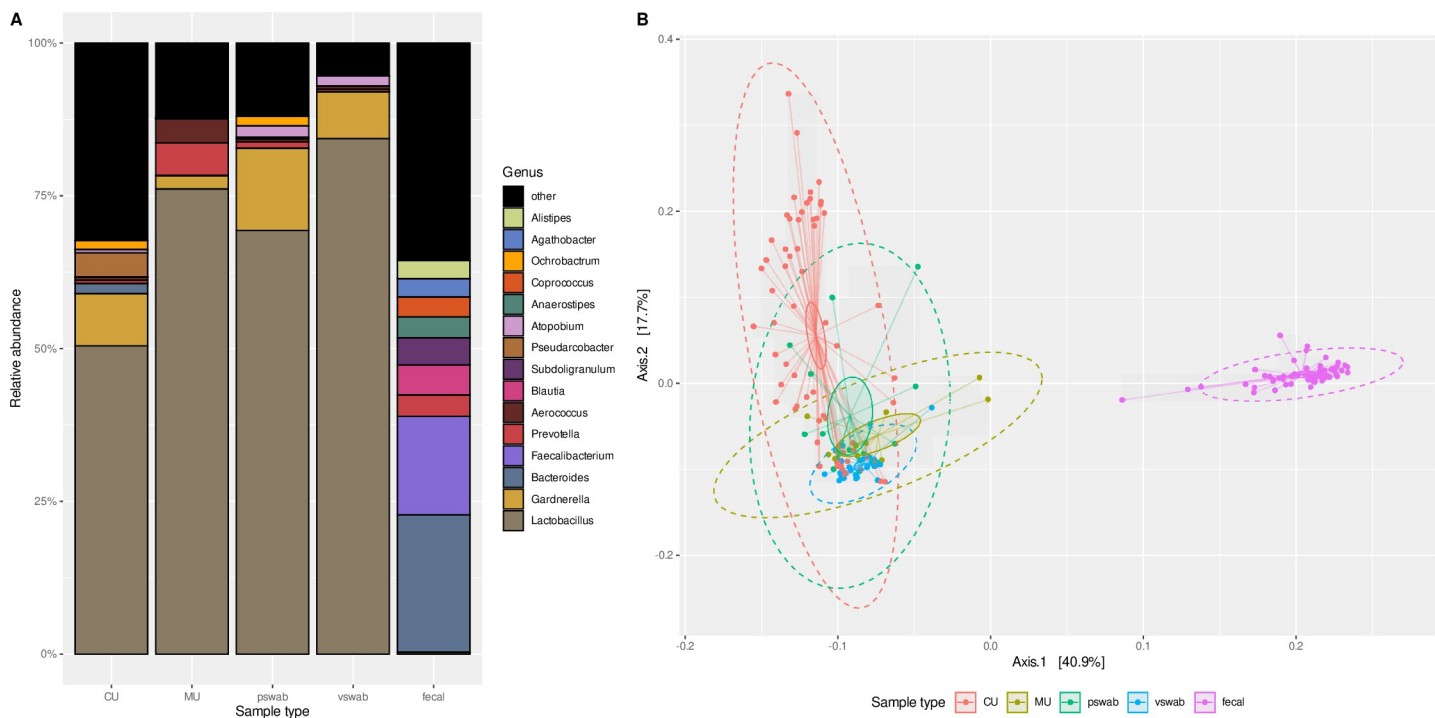

**Fig 2. Comparison of neighboring microbial habitats. A:** Fifteen most abundant taxa in all analyzed specimens (without UTI samples) shown with their mean relative abundance on genus level. **B:** Beta diversity (generalized UniFrac) of all analyzed samples in relation to sample type visualized by principal coordinates analysis (PCoA). 95% confidence levels assuming normal distribution (—) and 95% confidence ellipses (). Permutational multivariate analysis of variance using distance matrices (PERMANOVA) shows sample type as a significant factor explaining observed beta-diversity ($R^2$ = 0.48, p = 0.001). CU: Catheter urine; MU: Midstream urine; pswab: Periurethral swab, vswab: Vaginal swab.

**Table 3. Alpha diversity metrics of different samples types.**

| Sample type | Number of samples | Alpha diversity indices | | |
|---|---|---|---|---|
| | | Shannon Mean +/- SD | Inverse Simpson Mean +/- SD | PD Mean +/- SD |
| CU | 63 | 2.27 ± 1.34 | 10.67 ± 12.540 | 9.22±5.93 |
| MU | 10 | 1.06 ± 0.91 | 2.88 ± 3.34 | 4.38 ± 1.40 |
| Vswab | 30 | 0.65 ± 0.60 | 1.75 ± 1.15 | 2.94 ± 2.69 |
| Pswab | 15 | 1.31 ± 0.97 | 2.68 ± 2.28 | 6.71 ± 3.25 |
| Fecal sample | 59 | 3.91 ± 0.35 | 29.60 ± 10.15 | 7.84 ± 1.8 |

SD: Standard deviation; PD: Faith's phylogenetic diversity; CU: Catheter urine; MU: Midstream urine; Vswab: Vaginal swab; Pswab: Periurethral swab.

As a next step, we investigated community overlap between sampled body sites within volunteers. While we identified 20 ASVs which were present both in CU and fecal samples, for each "overlap pairs" at least one relative abundance (either the one in CU or in fecal sample) was below 0.2% making it difficult to judge whether the ASV was really present in both sites or the overlap rather observed due to index hopping or contamination (S1 File). Regarding overlap between CU, pswab and vswab, however, there was a more frequent and consistent overlap. The ASVs with the most frequent overlaps between CU, pswab and/or vswab belonged to *Lactobacillus* (ASV: f14141d03c23f3658014090c72c23866; 20 samples from 11 volunteers and ASV: bb48d4fe6403e7ecb4bc406617649cd8; 14 samples from 10 volunteers), Finegoldia (ASV: 3678d26eb8e3fd26e603acd051347085;7 samples from 6 volunteers) and Gardnerella (ASV: 4f17105bf002596e47d0a3d93d6f71e7; 7 samples from 4 volunteers). As seen in these numbers, there is an overlap on ASV level not only between sites within the same subject, but also between different subjects indicating a common urogenital microbiota in our cohort (see S2 File).

## CU and MU comparison

Six pairs of CU and MU samples collected during visit 5 were eligible for comparison of sequencing and culture results. As visible in Fig 1, some differences were seen in the microbiota composition. However, analysis of differential abundance of taxa with a minimum relative abundance above 1% showed only Peptoniphilus to be significantly more abundant in MU than corresponding CU samples (mean relative abundance and SD in MU 1.8%±3.2 vs. CU <0.1%±0.001; p = 0.031). See S4 Fig for box plots comparing all genera with a relative abundance >1%. Beta diversity analysis of CU and MU pairs showed that categorization by volunteer ID accounted for a substantial proportion of the observed beta diversity (PERMANOVA $R^2$ = 0.76, p = 0.004) while sample type (MU vs. CU) accounted for much less ($R^2$ = 0.09, p = 0.006). Furthermore, JSD values were calculated between the CU and MU samples of visit 5 of each volunteer and compared with corresponding JSD values from CU samples 1–4. With two exceptions (volunteer A and N), the divergence between the two samples from visit 5 was higher than that between CU samples collected during visits 1–4 (S5 Fig). Expanded culture yielded more species in 9/10 MU samples compared to the corresponding CU samples. In three cases, not all species cultured from CU were detected in the MU sample (S4 Table).

## UTI case

Between her third (day 117 after baseline) and fourth (day 215) study visit, participant K developed a symptomatic UTI with dysuria and hematuria and was prescribed a single dose of 3 g fosfomycin p.o. Expanded culture of a MU sample collected before antibiotic treatment (UTI1,

day 177) yielded *Escherichia coli*, *Streptococcus agalactiae*, *Staphylococcus lugdunensis*, *Propionibacterium avidum*, and *Actinomyces neuii*. Sequencing results of this and a CU sample collected three days later (UTI2) confirmed a clear shift in the composition of urinary microbiota with an emergence of the taxon *Escherichia-Shigella*. This taxon had not been present on genus level in her previous urine samples, neither in her last sample 38 days after the UTI, but was detected with a relative abundance of 3.4% in the fecal sample preceding the UTI (S6 Fig).

## Discussion

We investigated the longitudinal variability of urinary microbiota in healthy premenopausal women over a period of 6–18 months. Based on our results, the long-term stability of urinary microbiota compositions is lower than that observed in fecal microbiota. Furthermore, our analysis of both, catheterized and midstream urine collected in parallel, demonstrates some differences between the resulting microbiota compositions. Comprehensive beta diversity analyses confirmed a large overlap between urinary microbial communities and those in the vagina and periurethral region.

The presence of urinary microbiota could be confirmed in the majority of samples by both 16S rRNA gene sequencing (detection of sufficient bacterial DNA in 61/68 samples; 92.6%) and expanded culture (positive in 59/67 samples; 88.1%). Despite the use of a modified protocol for expanded culture, the proportion of culture-positive samples was slightly higher than previously reported proportions (51–80%) from studies using expanded quantitative urinary culture (EQUC) for CU samples [4,8,10]. The microbial composition of urinary samples from both culture and 16S rRNA gene sequencing, and the alpha diversity metrics are in line with previous findings with Lactobacillus and Gardnerella being the predominantly detected taxa [4,8–10]. As reported previously, our 16S rRNA gene sequencing showed low relative abundances of *Streptococcus* (max. 2.3% in one CU sample) even in subjects with culturally detectable *Streptococcus* showing that detection by our cultural procotol does not indicate high abundance in sequencing [9,10].

To our knowledge, this is the first study to assess the longitudinal variability of urinary microbiota in asymptomatic women by 16S rRNA gene sequencing of CU samples. Yet, a very recent study utilized EQUC of voided urine for the determination of urinary microbiota compositions and temporal changes with daily sample collection over a period of three months [16]. The authors observed notable and often transitional changes in the compositions of lower urinary tract microbiota, which could be attributed to menstruation and sexual intercourse. Both, this and our study utilized JSD as a measure of intraindividual variability with a lower JSD value representing higher similarity between compared samples. Despite methodological differences in study design, reported overall JSD values of voided urinary samples were similar to those in our study with CU samples, less frequent sampling and 16S rRNA gene sequencing instead of ECUQ [16]. The authors describe the urinary microbiota as both dynamic and resilient reflecting that some temporal changes in microbiota compositions were reversible [16]. In line with this observation, our beta diversity analysis including a PERMANOVA analysis showed that a significant proportion ($R^2$ = 0.48) of observed dissimilarities between CU samples could be explained by categorization by volunteer IDs. At least in some women, an underlying urinary microbiota signature seems to persist over the sampling period of six to eighteen months.

In our study, the intraindividual variability of urinary microbiota was compared to the values found in fecal samples from the same cohort. The overall range of JSD values was significantly lower in fecal samples indicating a longitudinally more stable microbial community in the gut. Likewise, categorization by volunteer ID accounted for a higher proportion of

observed beta diversity in fecal samples ($R^2$ = 0.59). This comparison is of particular interest for future research on the role of urinary microbiota in health and disease. While already many associations between specific alterations of the gut microbiota and disease have been well described, the higher intraindividual variability in healthy urinary microbiota likely complicates the establishment of similar associations for urinary microbiota.

In order to assess influence factors on compositional dynamics of urinary microbiota, we performed a multivariate analysis on beta diversity. Despite the small sample size, we were able to confirm a potential impact of time since last sexual intercourse ($R^2$ = 0.09) on urinary microbiota compositions in both models with and without stratification to volunteer ID. Recent sexual activity has been previously reported to influence urinary microbiota [16]. Furthermore, our analysis indicated a potential influence by daily fluid intake. While our study design with few samples per volunteer did not allow for a more robust assessment of factors influencing intraindividual dynamics of urinary microbiota, future studies should include assessment or control for these factors in their study design and should evaluate different approaches to influence factor analaysis [33].

In our sample collection, the most marked temporal changes in urinary microbiota were observed in one volunteer who developed a UTI during the study period. The respective urinary samples displayed an altered microbiota composition with an increase in the taxon *Escherichia/Shigella* and a decreased alpha diversity. Of note, we did not identify a corresponding alteration in the preceding urinary sample 60 days prior to the emergence of the UTI. However, the respective fecal sample yielded *Escherichia/Shigella* with a relative abundance of 3.4%, which could indicate a possible transition of the causative pathogen from the intestine to the bladder prior UTI emergence. Relative abundances of uropathogens over 1% have been described as a risk factor for UTI in kidney transplant patients [34], but the presence in the intestine observed in this single case could also be completely unrelated to the emergence of the UTI.

Comparison of microbiota found in CU samples with neighboring microbial communities in pswabs, vswabs, and fecal samples, showed a clear distinction between the microbial habitats in the intestine and the urogenital tract. The large overlap in most abundant taxa both on genus and ASV level, in particular the high proportion of the genus *Lactobacillus*, as well as the close proximity in beta diversity analyses, confirm the previously postulated interconnectedness of female urogenital microbiota [18,19,21]. However, within urogenital samples, differences between microbiota compositions derived from CU samples to those from MU samples have been identified [6,22,35]. For comparison of CU to MU samples in our study, voiding took place directly after catheterization. According to our beta diversity analysis, microbiota in MU samples were more closely related to those detected in vaginal and periurethral samples suggesting a larger proportion of urethral microbiota as described before [22]. In line with this finding, expanded culture detected a higher variety of bacterial species in MU as compared to CU samples. Furthermore, the divergence of microbiota compositions measured by JSD values was higher between pairs of CU and MU samples collected at the same visit than between the four CU samples collected within six months. Differences between CU and MU samples and implications for diagnostics of UTIs and utilization in microbiota research are a matter of debate [6,17,35]. The advantages of potentially lower contamination need to be weighed against higher invasiveness of transurethral catheterization. Additionally, depending on the urogenital condition under scrutiny, samples reflecting rather the urethral than the bladder microbiota may even be of higher interest. Of note, alterations associated with the UTI in one of our volunteers were seen in both, a MU and a CU sample, following diagnosis. Given the aforementioned close relatedness of microbial communities in the female urogenital tract, alterations and dynamics associated with disease might be represented in both specimen and pragmatical study designs relying rather on MU than CU samples seem justifiable.

Our study has some limitations. First, due to the invasiveness of our sampling technique, we collected only a very limited number of samples per patient with two months time between the main sampling visits. In two volunteers, one CU sampling each was missed. As a result, interim alterations of urinary microbiota may have been missed. Furthermore, this low number of samples together with the rather discrete changes in volunteer reported variables limit the reliability of our influence factor analysis. In particular, to study changes related to the menstrual cycle, more frequent sampling within one cycle would have been necessary. Second, our analysis of negative controls revealed a substantial contamination of CU samples with five repeatedly detected ASVs, which were most likely introduced during processing of samples. CU samples are particularly prone to contamination due to their low biomass and bacterial load [36]. While there is no consensus how to determine contamination and how to extract the respective sequences from the analysis, the negative correlation between DNA concentration of a sample and the abundance of these ASVs supports our interpretation as contaminants. Since both, sampling and extraction blanks yielded these ASVs, we suspect an introduction of the contamination by extraction reagents as previously described [37]. Unfortunately, the applied reagents were no more available after finalization of the bioinformatics interpretation. In addition, we did not do extraction replicates that would have allowed us to confirm the findings seen in these low-abundance samples. Third, our cohort was very homogenous with regard to age. As a consequence, age was not included in the PERMANOVA analysis despite being previously reported as influential [21,38]. While this may reduce the generalizability of our results, the homogeneity of the cohort probably provided us with the opportunity to detect similarities in longitudinal variability despite the small sample size. Fourth, an unexpectedly high proportion of volunteers (4/15) received antibiotic treatment during the study, which possibly influenced their urinary microbiota. However, the time span between antibiotic exposure and urinary sampling was long for most samples, and taxa bar plots showed no clear effect on urinary microbiota compositions (C, D and E) except for a clear change seen in the one volunteer that developed a UTI (K). While the impact of antibiotic exposure on the urinary microbiota warrants further investigation, we do not feel like the treatments reported in our cohort significantly affected our study findings. Fifth, we did not include any means to quantify bacterial load in the urinary sample and, thus, were not able to detect potential quantitative alterations of urinary microbiota. Sixth, DNA extraction methods differed for each sample type. This could impact the comparability of results between sample types, however, it allowed us to use extraction protocols already well validated for a certain sample type. Finally, we had to restrict expanded culture to urine samples and 16S rRNA gene sequencing to urine, fecal samples, and only a selection of pswabs and vswabs due to limited resources. By our focused culture approach, we were not able to compare cultured bacteria from the different urogenital reservoirs to confirm overlap at the bacterial strain level i.e. by means of whole-genome sequencing. So far, one study has demonstrated strain identity in CU and vaginal samples from four patients [18]. Further research at the strain level including periurethral, but also fecal samples and utilizing metagenomics approaches for microbiota analysis seems warranted. By not sequencing all pswabs and vswabs, our study cannot demonstrate longitudinal variability of the periurethral and vaginal microbiota. However, even with these constraints, our comprehensive sampling protocol provided us with the unique opportunity to study urogenital and fecal microbiota from identical individuals and, in one case, present alterations associated with the emergence of a UTI.

In conclusion, our study shows a longitudinally variable urinary microbiota in healthy premenopausal women over a period of six and, in a substudy, 18 months. The observed variability was higher than that observed in fecal samples of the same cohort. We confirmed an influence of time since last sexual intercourse on urinary microbiota compositions. Our results

add to the very recent evidence of closely connected microbial habitats in the urogenital tract while demonstrating rather discrete differences between microbiota derived from catheterized urine to those from midstream urine. The observation of longitudinal microbiota alterations associated with the emergence of a UTI highlights the need for further clinical studies investigating temporal microbiota dynamics of different reservoirs in urogenital diseases.

## Supporting information

**S1 Fig. Relative abundance of bacterial genera in control samples.**
(PDF)

**S2 Fig. Intraindividual changes in alpha diversity of urine samples.**
(PDF)

**S3 Fig. Beta-diversity of CU, MU, pswab and vswab visualized by PCoA.**
(PDF)

**S4 Fig. Comparison of CU and MU samples.**
(PDF)

**S5 Fig. Variability of CU samples and MU samples shown as median Jensen-Shannon Divergence values.**
(PDF)

**S6 Fig. Taxa abundance plots of volunteer K.**
(PDF)

**S1 Table. Fifteen most abundant taxa on genus level per sample type.**
(PDF)

**S2 Table. Most frequently detected species in expanded urinary culture of CU samples.**
(PDF)

**S3 Table. Comparison of urinary microbiota results of 16S rRNA sequencing and expanded culture of CU samples including those treated with PMA.**
(PDF)

**S4 Table. Comparison of cultural results from CU and corresponding MU samples.**
(PDF)

**S1 File. Relative abundances of overlapping ASVs in CU and fecal samples.**
(PDF)

**S2 File. Relative abundances of overlapping ASVs in CU, pswab and vswab samples.**
(PDF)

## Acknowledgments

We would like to thank Dr. Markus Valter for providing sample collection facilities, and all volunteers for participation. Thanks also to Fabiola Sack and Barbara Kneiding for laboratory support.

## Author Contributions

**Conceptualization:** Lena M. Biehl, Hilmar Wisplinghoff, Maria J. G. T. Vehreschild.

**Data curation:** Lena M. Biehl, Fedja Farowski, Catharina Hilpert.

**Formal analysis:** Fedja Farowski, Catharina Hilpert, Angela Nowag.

**Funding acquisition:** Lena M. Biehl.

**Investigation:** Lena M. Biehl, Catharina Hilpert, Angela Nowag, Anne Kretzschmar, Nathalie Jazmati, Anastasia Tsakmaklis, Yascha Khodamoradi.

**Methodology:** Lena M. Biehl, Angela Nowag, Anne Kretzschmar, Nathalie Jazmati, Anastasia Tsakmaklis, Imke Wieters, Hilmar Wisplinghoff.

**Project administration:** Lena M. Biehl, Catharina Hilpert, Hilmar Wisplinghoff, Maria J. G. T. Vehreschild.

**Resources:** Lena M. Biehl, Anastasia Tsakmaklis, Yascha Khodamoradi, Hilmar Wisplinghoff.

**Software:** Fedja Farowski.

**Supervision:** Lena M. Biehl, Maria J. G. T. Vehreschild.

**Validation:** Fedja Farowski, Imke Wieters.

**Visualization:** Fedja Farowski.

**Writing – original draft:** Lena M. Biehl, Fedja Farowski.

**Writing – review & editing:** Lena M. Biehl, Fedja Farowski, Catharina Hilpert, Angela Nowag, Anne Kretzschmar, Nathalie Jazmati, Anastasia Tsakmaklis, Imke Wieters, Yascha Khodamoradi, Hilmar Wisplinghoff, Maria J. G. T. Vehreschild.

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
