## [Decision Letter · Decision Letter 0]

17 May 2021

PONE-D-21-11667

Longitudinal variability in the urinary microbiota of healthy premenopausal women and the relation to neighboring microbial communities: a cohort study

PLOS ONE

Dear Dr. Vehreschild,

Thank you for submitting your manuscript to PLOS ONE. After careful consideration, we feel that it has merit but does not fully meet PLOS ONE’s publication criteria as it currently stands. Therefore, we invite you to submit a revised version of the manuscript that addresses the points raised during the review process.

We look forward to receiving your revised manuscript.

Kind regards,

John Richard Lee, M.D.

Academic Editor

PLOS ONE

Additional Editor Comments:

Please address the thoughtful comments and suggestions for improvement from the Reviewers.

Journal Requirements:

[I have read the journal's policy and the authors of this manuscript have the following competing interests: LMB has received lecture honoraria from Astellas and Merck/MSD, and travel grants from 3M and Gilead.

MJGTV reports grants and personal fees from 3M, Alb Fils Kliniken GmbH, Astellas Pharma, Basilea, bioMérieux, DaVolterra, Gilead Sciences, Ferring, Glycom, Heel, MaaT Pharma, Merck/MSD, Organobalance, Pfizer, Roche Pharma, Seres Therapeutics.

YK has received lecture honoraria from Merck/MSD and Gilead, and travel grants from Gilead.

All remaining authors have declared no conflicts of interest].

Reviewers' comments:

Reviewer's Responses to Questions

**Comments to the Author**

1. Is the manuscript technically sound, and do the data support the conclusions?

Reviewer #1: No

Reviewer #2: Yes

2. Has the statistical analysis been performed appropriately and rigorously? 

Reviewer #1: Yes

Reviewer #2: Yes

3. Have the authors made all data underlying the findings in their manuscript fully available?

Reviewer #1: Yes

Reviewer #2: Yes

4. Is the manuscript presented in an intelligible fashion and written in standard English?

Reviewer #1: Yes

Reviewer #2: Yes

5. Review Comments to the Author

Reviewer #1: Longitudinal variability in the urinary microbiota of healthy premenopausal women and the relation to neighboring microbial communities: a cohort study

Revisions

The authors have submitted a well written manuscript, describing an interesting study. Studies on the urinary microbiota are being published, but so far, they have mostly been of cross-sectional design. Longitudinal studies are highly warranted.

I do, however, have a few comments on the manuscript.

Title

I would suggest changing the study description from “a cohort study” to “a pilot study” considering the low number of included participants.

Introduction

L. 72-73: please put in the reference.

Methods

L. 148: please provide name of manufacturer.

L. 152-160: Have replicate DNA extractions been made or only one extraction per participant? Use of replicates would strengthen the study, especially when analyzing samples with low DNA concentrations and sample numbers are low.

Results

L. 237-238: How did antibiotic treatment affect the results? And what was the time span from treatment to sample collection.

L. 286-287: Were there any significant changes in alpha-diversity? It is very difficult to see anything from that figure.

L. 287-289: The authors write that the microbiota composition remained similar throughout the study for most volunteers. I disagree with this statement, since several of the participants show shifts in urotypes over the collection period. For instance, there are shifts between bacteroides urotypes and lactobacillus urotypes for participant A, B, I, J, K, L, M and O.

Discussion

L. 407-409: Again, I disagree with the conclusion on a long-term stability of the urinary microbiota. The composition appears to shift a great deal.

L. 430-432: The authors note that, unlike others, they find a high proportion of urine samples with high relative abundances of Bacteroides. This is very puzzling, and I am concerned that cross-contamination – either during sampling or the following lab analyses, occurred. The authors claim that it is highly unlikely that Bacteroides was introduced as a laboratory contaminant and that the negative controls were instead contaminated by urine samples. I find this less likely. Instead the use of increased input volumes of extracted DNA, as described in the methods section, could increase contamination contribution from sampling or extraction procedures. Do they see an overlap between samples with higher input volumes and Bacteroides urotypes? Could the authors please provide sequencing data on the negative controls and further describe how the negative controls were constructed and handled? This is lacking in the methods section.

Furthermore, the authors speculate that the high degree of Bacteroides could be specific for premenopausal women. However, two studies (Price et al. 2019. The urobiome of continent adult women: a cross-sectional study and Amitzbøll et al., 2021. Pre- and postmenopausal women have different core urinary microbiota), have investigated the urinary microbiota (catheterized) of premenopausal women, and did not identify Bacteroides as a main contributor. The authors should include a discussion of their results against these two studies.

L. 438: The authors have recorded the time of menstruation for their participants, but do not report on how this might affect their results. Does collection of samples always occur at the same time in the menstrual cycle for all women? Or does it vary for the individual participants and also between participants?

Reviewer #2: Summary

In this submission, Vehreschild and colleagues report the stability of the urinary microbiome in healthy premenopausal women. The study is well-done and addresses a clinically useful topic with long-term clinical consequences. Prior to publication, I have a number of questions and concerns that should be addressed.

Major

-The biggest methodological limitation in this study is the high proportion of missed samples, patients with unexpected antibiotics, and infections. While this is unavoidable in a cohort study, it limits the conclusions that can be drawn from an already small sample size. This should be more clearly stated as a limitation.

-The authors should consider performing metagenomic analysis, even if only on a subset of samples, in future studies to assess strain-level taxonomic resolution and to definitively clarify whether strains in feces and urine overlap.

Title

-Appropriate

Abstract

-Appropriate

Intro

-Line 88 – this may be true for the female microbiome, but there is evidence in males (e.g. PMID 30143471). This should be cited or re-worded.

Methods

-As the authors acknowledge, the difference in sample processing methodology between samples of varying sources could add significant confounding effects on the comparison of data between them. Ideally, a standardized protocol should have been developed, but this is now beyond the scope of this manuscript.

-Were women with inflammatory bowel disease included?

-What taxa were identified in negative extraction and PCR controls? Were these subtracted out from specimens?

-The “week of menstrual cycle” component of this study is an admirable attempt at capturing this, but the small number of samples per patient weakens this. I suspect that large differences occur over the short period of time when menstruation is dynamically changing and a variety of products are being utilized.

-Was ethnicity captured in this data set?

Results

-Clarify whether the patient with MS had any neurogenic bladder symptoms or workup

-The high rate of infection and use of antibiotics by participants during this study is a potential limitation and should be stated as such.

-Why were ~30% of EQUC isolates not seen on 16s? Is this a classification error? Is there better correlation at genus/family/order levels?

-Time since last intercourse is a known variable affecting the microbiome. The authors should expand upon this in the discussion as a key component of the study.

Discussion

-More time could be spent discussing how an individual’s microbiome can be so different than other healthy volunteers and yet stable and non-pathogenic. Does this imply a large degree of tolerability in the female GU microbiome? It certainly will make pathogenic characteristics more difficult to identify.

-The PMA discussion section could be significantly shortened

References

-I would recommend that the authors update their literature search, as several urinary microbiome studies have been recently published that warrant consideration and/or discussion

Figures

-The low resolution nature of the figures makes them illegible for review. This makes review of the manuscript very difficult.

6. PLOS authors have the option to publish the peer review history of their article (what does this mean?). If published, this will include your full peer review and any attached files.

Reviewer #1: No

Reviewer #2: No

---

## [Author Response · Author response to Decision Letter 0]

1 Oct 2021

We would like to thank you and the reviewers for the helpful and constructive comments. We were able to reply to all comments and address nearly all suggestions in our revised version of the manuscript. Please see file Responsse to reviewers for a point-by-point reply to all reviewers' comments.

---

## [Decision Letter · Decision Letter 1]

3 Nov 2021

PONE-D-21-11667R1Longitudinal variability in the urinary microbiota of healthy premenopausal women and the relation to neighboring microbial communities: a pilot studyPLOS ONE

Dear Dr. Vehreschild,

Thank you for submitting your manuscript to PLOS ONE. After careful consideration, we feel that it has merit but does not fully meet PLOS ONE’s publication criteria as it currently stands. Therefore, we invite you to submit a revised version of the manuscript that addresses the points raised during the review process.

We look forward to receiving your revised manuscript.

Kind regards,

John Richard Lee, M.D.

Academic Editor

PLOS ONE

Journal Requirements:

Additional Editor Comments:

Thank you for submitting the revised manuscript which has been significantly improved. Please address the last comments of the authors. Please explain/justify the use of the stepwise multivariate analysis. Also please explain which variables went into the model and how it was conducted in more detail.

Reviewers' comments:

Reviewer's Responses to Questions

**Comments to the Author**

1. If the authors have adequately addressed your comments raised in a previous round of review and you feel that this manuscript is now acceptable for publication, you may indicate that here to bypass the “Comments to the Author” section, enter your conflict of interest statement in the “Confidential to Editor” section, and submit your "Accept" recommendation.

Reviewer #1: (No Response)

Reviewer #2: All comments have been addressed

2. Is the manuscript technically sound, and do the data support the conclusions?

Reviewer #1: Yes

Reviewer #2: Yes

3. Has the statistical analysis been performed appropriately and rigorously? 

Reviewer #1: Yes

Reviewer #2: I Don't Know

4. Have the authors made all data underlying the findings in their manuscript fully available?

Reviewer #1: Yes

Reviewer #2: Yes

5. Is the manuscript presented in an intelligible fashion and written in standard English?

Reviewer #1: Yes

Reviewer #2: Yes

6. Review Comments to the Author

Reviewer #1: Thank you for your very thorough response and the choice to go more into the data of the negative controls.

I still have one comment:

In line 315-318 you write that there are only few intra-individual chages in alpha-diversity (S2), and furthermore that the relative abundance of taxa remain similar throughout the study. I think that you have fogotten to update this section according to your new analyses.

Reviewer #2: The authors are to be commended on an improved manuscript now worthy of publication. A few minor concerns remain:

The use of stepwise multivariate models is controversial and should be justified if not validated

Vaughan et al (PMID 34181466) should be cited

Line 168 – lysozyme is missing the “e”

Line 385 – unclear if the “fig” is supposed to be there

Line 400 – Visible is mis-spelled

Line 402 – peptoniphilus is mis-spelled

7. PLOS authors have the option to publish the peer review history of their article (what does this mean?). If published, this will include your full peer review and any attached files.

Reviewer #1: No

Reviewer #2: No

---

## [Author Response · Author response to Decision Letter 1]

15 Dec 2021

Thank you for your helpful remarks and suggestions. Please see the file "Response to reviewers" for our detailled reply.

---

## [Editor Report · Decision Letter 2]

19 Dec 2021

Longitudinal variability in the urinary microbiota of healthy premenopausal women and the relation to neighboring microbial communities: a pilot study

PONE-D-21-11667R2

Dear Dr. Vehreschild,

We’re pleased to inform you that your manuscript has been judged scientifically suitable for publication and will be formally accepted for publication once it meets all outstanding technical requirements.

Kind regards,

John Richard Lee, M.D.

Academic Editor

PLOS ONE
---

## [Editor Report · Acceptance letter]

6 Jan 2022

PONE-D-21-11667R2 

Longitudinal variability in the urinary microbiota of healthy premenopausal women and the relation to neighboring microbial communities: a pilot study 

Dear Dr. Vehreschild:

I'm pleased to inform you that your manuscript has been deemed suitable for publication in PLOS ONE. Congratulations! Your manuscript is now with our production department. 

Kind regards, 

on behalf of

Dr. John Richard Lee 

Academic Editor

PLOS ONE